# Assessing the Mass Sensitivity for Different Electrode Materials Commonly Used in Quartz Crystal Microbalances (QCMs)

**DOI:** 10.3390/s19183968

**Published:** 2019-09-14

**Authors:** Xianhe Huang, Qiao Chen, Wei Pan, Jianguo Hu, Yao Yao

**Affiliations:** 1School of Automation Engineering, University of Electronic Science and Technology of China, Chengdu 611731, China; qiaochen@std.uestc.edu.cn(Q.C.); weipan@std.uestc.edu.cn(W.P.); hujianguo@std.uestc.edu.cn(J.H.); yaoyao428@uestc.edu.cn(Y.Y.); 2Institut für Informatik VI, Technische Universität München, Schleißheimer Straße 90a, Garching 85748, Germany

**Keywords:** quartz crystal microbalance (QCM), mass sensitivity, electrode materials, thin film

## Abstract

Mass sensitivity is vital for quartz crystal microbalance (QCM)-based data analysis. The mass sensitivity distribution of QCMs may differ greatly depending on the shapes, thicknesses, sizes, and materials of the metal electrodes. This is not considered by the Sauerbrey equation, and has a large potential to cause errors in QCM-based data analysis. Many previous works have studied the effects of shape, thickness, and size of metal electrodes on mass sensitivity. However, it is necessary to continue to clarify the relationship between the mass sensitivity and the electrode material of the QCM. In this paper, the results of both theoretical calculation and experimental analysis showed that the mass sensitivity of QCMs with gold electrodes is higher than that of the QCMs with silver electrodes, which in turn indicated that the mass sensitivity of QCMs varies with the electrode material. Meanwhile, the results of this study showed that the mass sensitivity of QCMs with different electrode materials is not proportional to the density of the electrode materials. This result suggests that, in order to obtain more accurate results in the practical applications of QCMs, the influence of electrode material on the mass sensitivity of the QCMs must be considered.

## 1. Introduction

The advent of quartz crystal microbalances (QCMs) has allowed intersectional research between piezoelectric devices and ultra-small mass sensing. QCMs have attracted increasing attention because of their simplicity, ease of use, low-cost components, and high sensitivity [1,2].

Because QCMs can accurately measure ultra-small masses in real time, they are used by researchers to study and observe extremely subtle changes in mass. During the past few decades, QCMs have been used in various technical and research applications, such as the monitoring of surface interaction processes [3,4,5], piezoelectric immunosensors [6,7,8], nanoscale characterization tools [9,10,11,12], and various specific molecule sensors [13,14].

Mass sensitivity is vital for QCM-based data analysis. Sauerbrey put forward the famous Sauerbrey equation, which describes the mass–frequency relationship of the surface of the QCM [15]:
(1)Δm=−2f02A(ρqμq)12*Δf=−CQCM*Δf
where ∆*m* and Δf are the mass change and frequency shift, respectively; *C_QCM_* is the mass sensitivity constant; $f_{0}$ is the fundamental frequency of the QCM; *A* is the effective area of the QCM; and $\rho_{q}$ and $\mu_{q}$ are the density and shear modulus of the piezoelectric quartz crystal, respectively.

Obviously, the Sauerbrey equation does not consider the effect of the electrodes. Consequently, when analyzing data, many early researchers used the same mass sensitivity values for QCMs with the same fundamental frequency but different electrode materials, e.g., the mass sensitivities of 5 MHz QCMs with gold and silver electrodes, respectively (labeled as Au-QCMs and Ag-QCMs, respectively) were both taken as 17.7 $\mathrm{ng}\cdot {\mathrm{cm}}^{-2}\cdot {\mathrm{Hz}}^{-1}$[16,17,18,19,20]. However, later, researchers gradually realized that the mass sensitivity of QCMs has a Gaussian distribution, rather than having the same mass sensitivity on the whole surface of the microbalance. Furthermore, the size and shape of the electrodes affect the mass sensitivity distribution of a QCM [21,22,23].

Our previous works [24,25,26,27] quantitatively analyzed the effects of the shape, thickness, and size of electrodes on the mass sensitivity of QCMs, while our latest paper [28] studied the relationship between the mass sensitivity of the n-electrode surface and the m-electrode surface in an n–m type QCM. 

It is worth noting that the cost of a QCM is positively related to the cost of the electrode material, with the prices of different electrode materials varying greatly. Once QCMs are commercialized and mass-produced, manufacturers will inevitably coat the surface of QCM electrodes with different materials (e.g., silver) to reduce the production costs. Accordingly, it is necessary to clarify the relationship between the mass sensitivity of QCMs and the types of electrode material in order to allow accurate QCM-based data analysis.

In this paper, taking Au-QCMs and Ag-QCMs as examples, we continue to study whether the mass sensitivity of QCMs is the same for different electrode materials, and, if it is different, what is the relationship between the QCM’s mass sensitivity and the density of the electrode material.

## 2. Theory

The mass sensitivity distribution on the electrode surface is not uniform, but rather has a Gaussian profile. The maximum mass sensitivity occurs at the center of the electrode, and this maximum mass sensitivity is referred to as the absolute mass sensitivity [29]. To a certain extent, the absolute mass sensitivity can reflect the sensitivity of the QCM. The mass sensitivity distribution function $S_{f}\left(r\right)$ is as follows [23,30]:(2)$$S_{f}\left(r\right)=\frac{{\left|A\left(r\right)\right|}^{2}}{2\pi \int_{0}^{\infty }r{\left|A\left(r\right)\right|}^{2}dr}\cdot C_{f}$$
where *r* is the distance from the electrode center, *A*(*r*) is the particle displacement amplitude function, and $C_{f}$ is Sauerbrey’s sensitivity constant. 

*A*(*r*) is the solution of the following Bessel equation [25,31]:(3)$$r^{2}\frac{\partial^{2}A}{\partial r^{2}}+r\frac{\partial A}{\partial r}+\frac{k_{i}^{2}r^{2}}{N}A=0$$
where N depends on the material constants of the quartz crystal; $k_{i}^{2}=\left(\omega^{2}-\omega_{i}^{2}\right)/c^{2}$, where i = *E*, *U* (*E* and *U* represent the fully electroded region and non-electroded region, respectively); $c=\sqrt{c_{66}/\rho_{q}}$ is the acoustic wave velocity in the crystal (where $c_{66}$ is the elastic stiffness constant and $\rho_{q}$ is the density of the quartz); and $\omega_{i}$ is the cut-off frequency of the fully electroded region ($\omega_{E}$) and the non-electroded region ($\omega_{U}$). 

Substituting the particle displacement amplitude into Equation (2), the mass sensitivity function of the QCM and its distribution can be obtained in order to analyze the influence of the electrode material on the mass sensitivity. Take AT-cut, “plano-plano” 10 MHz QCMs with an electrode diameter of 4 mm as an example. The theoretical absolute mass sensitivities of an Au-QCM and an Ag-QCM are 3.84 and 3.11 Hz·ng^−1^, respectively. That is, the mass sensitivity of QCMs vary with different electrode materials. Moreover, theoretical numerical results indicate that the mass sensitivity of QCMs is not proportional to the density of the electrode material.

## 3. Experiment

The results of theoretical calculation and analysis of QCM mass sensitivity show that the electrode material influences the mass sensitivity. In order to verify this result, it is necessary to measure the mass sensitivity of QCMs. However, it is difficult to directly detect the mass sensitivity of QCMs. Therefore, our previous work [29] proposed the equivalent mass sensitivity model, which takes into account the Gaussian distributional characteristics of QCM mass sensitivity and the effect of the electrode material on the mass sensitivity: (4)$$\Delta f=-C_{QCM}^{*}\times \Delta m\;\left(\mathrm{where}\;C_{QCM}^{*}=\frac{1}{\pi r_{d}^{2}}\int_{0}^{r_{d}}2\pi rS_{f}\left(r\right)dr\right)$$
where *r_d_* is the radius of the specified circular region onto which the mass load is attached, and $C_{QCM}^{*}$ is the equivalent mass sensitivity. The Sauerbrey equation is only applicable to thin, uniform, and rigid films. In this paper, thin and uniform mass films are the main subject of study, while discrete particle films or viscoelastic films are not considered.

The condition of the Sauerbrey equation can be better fulfilled by a method of plating thin, uniform rigid films onto the surface of QCM electrodes. A change in mass sensitivity can be verified by comparing the change in QCM frequency before and after the film coating. The present study used this method of comparing mass sensitivity to indirectly verify that the mass sensitivity of Au-QCMs is higher than that of Ag-QCMs.

The experiment was performed in a class 10,000 ultra-clean room at Wintron Electronic Co., Ltd. (Zhengzhou, China). The temperature and relative humidity of this room were kept constant at 23 °C(±2 °C)and 40% (±5%), respectively. For the experiment, the fundamental frequencies of all 20 AT-cut, plano-plano quartz wafers were 10 MHz, and the diameters of the wafers were 8.7 mm.

A schematic diagram of the experimental setup is shown in Figure 1. The 20 QCMs were evenly divided into two groups, namely an Au-QCM group and an Ag-QCM group, according to their electrode material.

Two S&A W-5600 base plating systems (Saunders & Associates, LLC, Phoenix, AZ, USA) were used in this experiment, one for gold plating and the other for silver plating. The thicknesses of the metal films were monitored using an INFICON SQC-310 deposition controller (East Syracuse, New York). In all plating processes, the vacuum pressure was less than $5\times 10^{-3}\;\mathrm{Pa}$and the evaporation rate of metal was set to 10 Å/s.

In the first plating process, the electrodes of the Au-QCM group were coated with gold, and those of the Ag-QCM group were coated with silver. The thickness and diameter of all electrodes were 1000 Å and 4 mm, respectively. The resonant frequencies of the QCMs were measured and recorded as $f_{1}$.

In the second plating process, the 10 QCMs in the Au-QCM group were divided into two subgroups, A and B, and the 10 QCMs in the Ag-QCM group were divided into two subgroups, C and D. In subgroup A, the electrodes of the five QCMs were coated with gold film, while in subgroup B, the electrodes of the five QCMs were coated with silver film. In subgroup C, the electrodes of five QCMs were coated with gold film, while in subgroup D, the electrodes of the five QCMs were coated with silver film. The diameter and thickness of all films in the second plating process were 1 mm and 500 Å, respectively. The resonant frequencies of QCMs were measured and recorded as $f_{2}.$

## 4. Results and Discussion

An S&A250B-1 network analyzer (Saunders & Associates, LLC, Phoenix, AZ, USA) was used to measure the frequencies of all 20 QCMs. All frequencies were recorded in air, and the results are shown in Table 1. In this experiment, the frequency shifts because of changes in temperature and humidity were almost negligible compared to the change in frequency caused by the mass change of the QCM surface.

The frequency shift, $\Delta f=f_{1}-f_{2},$ is caused by the mass change in the second plating process. $\bar{\Delta f}$ and δ are the average value and the standard deviation of the frequency shift in each subgroup, respectively. The mass of the gold and silver films, Δm, can be theoretically obtained from the density and volume of the films; here, the masses of the gold film and silver film were 757.91 and 412.33 ng, respectively. According to Equation (4), in this experiment, the equivalent mass sensitivities of the Au-QCMs and Ag-QCMs within the electrode central diameter of 1 mm were 3.64 and 2.97 Hz·ng^−1^, respectively. 

The theoretical frequency shift, $\Delta f_{e}$, was calculated according to the corresponding equivalent mass sensitivity and Δm. E_s_ is the deviation between $\bar{\Delta f}$ and $\Delta f_{e}$. The low standard deviation, δ, which was obtained for each subgroup, showed that the experimental system and environment were stable, and therefore showed that the experimental results are valid.

As can be seen from Table 1, the absolute values of the differences between the theoretical and experimental mass sensitivities of the Au-QCMs and Ag-QCMs were all less than 8.86%. Therefore, the experimental data can be considered to be in accordance with the theoretical results.

For subgroups A and C, the frequency shifts were different, while the mass changes caused by the second plating process were equal; the same was true for subgroups B and D. When a higher frequency shift is caused by the same mass change, it reflects a higher mass sensitivity. Thus, the mass sensitivity of the Au-QCMs is higher than that of the Ag-QCMs, which was consistent with theoretical calculations. That is, the electrode materials influenced the frequency–mass relationships of the QCMs, even though the Sauerbrey equation does not consider such an influence.

Additionally, the mass sensitivity of the Au-QCMs is about 1.1 times that of the Ag-QCMs, which is not proportional to the ratio between the densities of gold and silver. That is, the difference in the mass sensitivities of QCMs with different electrode materials is not proportional to the difference in the densities of the electrode materials. Our following work will investigate the deeper reason for the difference in mass sensitivity between QCMs with different electrode materials. 

## 5. Conclusions

In this paper, the results of both theoretical calculation and experiment showed that the mass sensitivity of Au-QCMs is higher than that of Ag-QCMs, which indicated that the mass sensitivity of QCMs varies with the electrode material. Furthermore, the results showed that the difference in mass sensitivity of QCMs with different electrode materials is not proportional to the difference in the density of the electrode material. This result suggests that, in order to obtain more accurate results in practical applications of QCMs, the influence of electrode material on the mass sensitivity of the QCM must be considered.

## Figures and Tables

**Figure 1 sensors-19-03968-f001:**
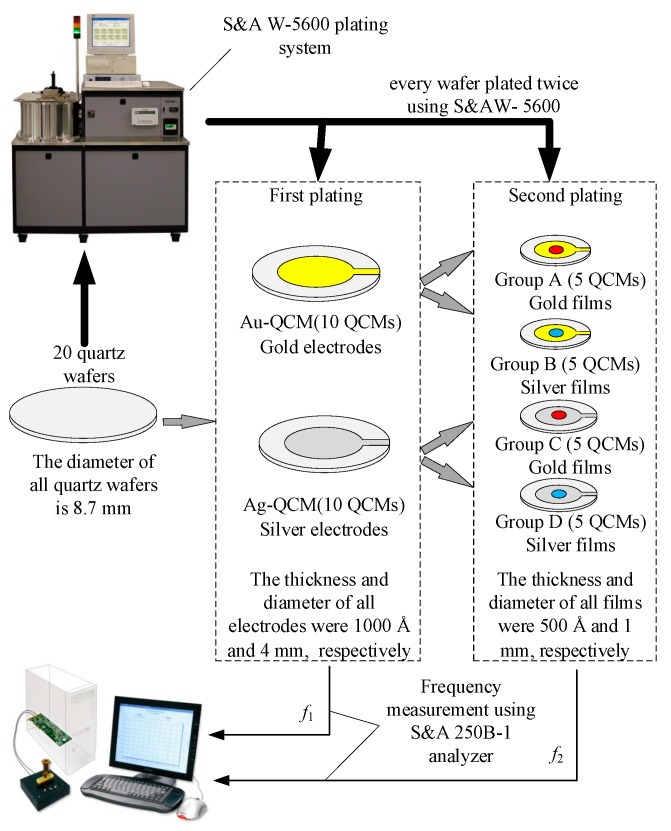
Schematic diagram of the experimental setup.

**Table 1 sensors-19-03968-t001:** Results of the quartz crystal microbalance (QCM) plating experiments.

Groups		$f_{1}\left(\mathrm{Hz}\right)$	$f_{2}\left(\mathrm{Hz}\right)$	Δf(Hz)	$\bar{\Delta f}\left(\mathrm{Hz}\right)$	δ(Hz)	Δm(ng)	$\Delta f_{e}\left(\mathrm{Hz}\right)$	*Es*
Au-QCM	A	9,961,970	9,959,360	2610	2662	69.79	757.91	2759	3.52%
		9,963,680	9,960,970	2710					
		9,963,990	9,961,360	2630					
		9,962,450	9,959,850	2600					
		9,963,490	9,960,730	2760					
	B	9,962,620	9,961,290	1330	1368	50.20	412.33	1501	8.86%
		9,962,550	9,961,220	1330					
		9,962,080	9,960,730	1350					
		9,962,740	9,961,290	1450					
		9,963,580	9,962,200	1380					
Ag-QCM	C	10,000,940	9,998,460	2480	2428	70.90	757.91	2251	7.86%
		10,002,010	9,999,590	2420					
		10,001,480	9,998,960	2520					
		9,999,510	9,997,140	2370					
		9,997,960	9,995,610	2350					
	D	10,001,480	10,000,270	1210	1222	19.24	412.33	1225	0.02%
		10,001,300	10,000,080	1220					
		10,001,650	10,000,450	1200					
		9,999,620	9,998,390	1230					
		9,995,910	9,994,660	1250					

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
