# Peer review of "Assessing the Mass Sensitivity for Different Electrode Materials Commonly Used in Quartz Crystal Microbalances (QCMs)"

_sensors, 2019, doi:10.3390/s19183968_

Round 1

Reviewer 1 Report

The authors reported different electrodes on the mass sensitivities of QCM technique. This is an interesting and important research topic for further exploration of QCM sensing methods. It can be accepted after addressing following aspects.

The experimental details of Au and Ag deposition should be specified. As this manuscript compares the differences of mass sensitivities of Au and Ag on QCM, the thicknesses of these two coatings should be exactly the same. How is the thickness monitored? What are the conditions when recording the frequencies of the QCM crystals? In vacuum or atmosphere? This is also very important, as the frequency is also sensitive to the measurement environment. The authors found the mass sensitivity between gold are less than the density difference, and they did not give any scientific input for the origin of these differences. The referee consider it might related to the differences of electrical conductivities. As the author just compared two different materials, the referee highly recommend the authors to investigate more electrode materials. It will not take much time. By doing so, it may give an indication of possible reasons for the differences of mass sensitivities. Some minor aspects: 1) in figure 1, the second one in the right column should be “group B”; 2) in page 3, line 115, the unit of mass sensitivity is Hz/ng or ng/Hz?

Author Response

thanks for your time to comment our paper, and our reply is shown in attachment.

Reviewer 2 Report

I´m a QCM user since many years ago and I´ve always been fascinated for the versatility of this instrument. Most commercial QCMs (i.e., Q-sense) use a 4.95 MHz AT-cut quartz crystal gold coated. Yet, some authors have reported is possible and electrodeless approach (Beck et al, Phys. Chem. Chem. Phys. 1988, 92, 1363–1368; Ogi et al, M. Anal. Chem. 2006, 78, 6903–6909). Could the authors emphasize on why the need of using different materials to surface coat the QCM electrodes if there is a mass sensitivity effect?

Could the authors comment on the applicability or limitations of their model to improve the mass sensitivity of different films deposited on the electrode surface: (i) thin and homogeneous rigid films; (ii) films consisting of discrete particles (i.e., viruses, micelles) that dissipate little energy, (iii) dissipating films due to the viscoelastic properties of the material?

Clarity and homogeneity in this manuscript will improve upon careful proofread. Although I was able to go through the text, I had a hard time following some of the ideas.  

Author Response

thanks for your attention to comment our paper, and our reply is shown in attachment.

Round 2

Reviewer 1 Report

The authors mentioned that the thickness was controlled by instrument. However, it is not clear how the thickness controlled. It is written on the specifications of the INFICON SQC-310 Deposition Controller that the thickness monitoring is based QCM measurement. However, in this study, the authors claimed the sensitivity difference between different electrodes, and the Sauerbrey equation is not directly applicable for mass determination. So if it is that case, the thickness control for the second layer deposition might not precise, or even on the first layer, because the thickness control might not be linear in the case of Ag on Au electrodes. In terms of the frequency recording, the measurement in cleanroom does not guarantee the same environment among different samples, because QCM is a very sensitive tool, slightly difference of temperature or humidity will give different results on the same sample. Therefore, it is recommended to do the measurement under controlled environment. It was recommended to study more materials to gain more insights on the difference. The excuse of time consuming for not doing this is not acceptable for scientific studies. And I do not believe this kind of experiments will take much time.

Author Response

Thanks for your comment.

This manuscript has been edited by native speakers who help to improve grammar and phrasing in MDPI English editing service.

Our response are shown in attachment.

Reviewer 2 Report

In my opinion the authors have adequately addressed my comments and suggestions. The only concern I have is the clarity of the manuscript.

I include a few list of examples:

*Lines 2-3 - Consider a more specific title , e.g. "Assessing the mass sensitivity for different electrode materials commonly used in quartz crystal microbalances (QCMs)".

Some sentences could be shorten or reworded:

*Lines 36-37 - "Sauerbrey put forward the famous Sauerbrey equation which describes the mass-frequency relationship of the surface of the QCM [15]"

*Lines 62-65 - "The mass sensitivity distribution on the electrode surface is not uniform but gaussian profile, the maximum mass sensitivity is at the center of the electrode and this maximum mass sensitivity is referred to as absolute mass sensitivity [29], which could to a certain extent reflect the sensitivity level of QCM."

*Through the text there are also several propositions that haven't been used correctly. 

Author Response

Thanks for your comment. our responses are shown in atttchment.

Round 3

Reviewer 1 Report

The revised version and the authors' elaboration are much clear in the revised version. I think it can be accepted after carefully polishing the English writing. Here just type some issues:

"works" should be "work" In some places, past tense should be used instead of present tense.

Author Response

 Thanks for your comment. On the basis of the English-edited version (09 September, 2019) by native English-speaking editors, we checked the English writing as well as the tense in the whole article again, and made some related revisions. Revised portion are marked in yellow in the paper.